# An Overview of IoT Sensor Data Processing, Fusion, and Analysis Techniques

**DOI:** 10.3390/s20216076

**Published:** 2020-10-26

**Authors:** Rajalakshmi Krishnamurthi, Adarsh Kumar, Dhanalekshmi Gopinathan, Anand Nayyar, Basit Qureshi

**Affiliations:** 1Department of Computer Science and Engineering, Jaypee Institute of Information Technology, Noida 201309, India; k.rajalakshmi@jiit.ac.in (R.K.); dhanalekshmi.g@jiit.ac.in (D.G.); 2School of Computer Science, University of Petroleum and Energy Studies, Dehradun 248007, India; adarsh.kumar@ddn.upes.ac.in; 3Graduate School, Duy Tan University, Da Nang 550000, Vietnam; 4Faculty of Information Technology, Duy Tan University, Da Nang 550000, Vietnam; 5Department of Computer Science, Prince Sultan University, Riyadh 11586, Saudi Arabia; qureshi@psu.edu.sa

**Keywords:** Internet of Things, data processing, data analysis, data fusion, emerging technologies

## Abstract

In the recent era of the Internet of Things, the dominant role of sensors and the Internet provides a solution to a wide variety of real-life problems. Such applications include smart city, smart healthcare systems, smart building, smart transport and smart environment. However, the real-time IoT sensor data include several challenges, such as a deluge of unclean sensor data and a high resource-consumption cost. As such, this paper addresses how to process IoT sensor data, fusion with other data sources, and analyses to produce knowledgeable insight into hidden data patterns for rapid decision-making. This paper addresses the data processing techniques such as data denoising, data outlier detection, missing data imputation and data aggregation. Further, it elaborates on the necessity of data fusion and various data fusion methods such as direct fusion, associated feature extraction, and identity declaration data fusion. This paper also aims to address data analysis integration with emerging technologies, such as cloud computing, fog computing and edge computing, towards various challenges in IoT sensor network and sensor data analysis. In summary, this paper is the first of its kind to present a complete overview of IoT sensor data processing, fusion and analysis techniques.

## 1. Introduction

In the coming years of the Internet of Things (IoT), context-awareness bridges the interconnection between the physical world and virtual computing entities, and involves environment sensing, network communication, and data analysis methodologies [1]. Advancement enables several advanced IoT applications, such as intelligent healthcare systems, smart transport systems, smart energy systems and smart buildings. The IoT networks’ unified architecture includes smart IoT-based application services and the underlying IoT sensor networks [2]. According to the Gartner forecast, the IoT global market envisions 5.8 billion IoT-based applications by 2020, with a 21% increase from 2019 [3]. Further, the IoT market’s worldwide growth is propelled by wireless networking technologies and the adoption of emerging technologies such as cloud platforms. This trend leads to a drastic increase in demand for connected IoT devices and application services.

The primary objectives of IoT sensor networks include (i) sensing the critical information from the external physical environment, (ii) the sampling of internal system signals, and (iii) obtaining meaningful information from sensor data to perform decision-making [4,5]. It is to be noted that IoT-enabled applications involve a wireless sensor network (WSN). Further, these wireless sensors are randomly positioned and capable of establishing an ad hoc network without infrastructure requirements. The wireless sensor network is reinforced by low-cost and lower power devices, such as Wi-Fi, Bluetooth, Zigbee, Near Frequency Communication, etc. However, these wireless-based networks incur difficulties, such as inference, loss of data, redundancy of data and different data generation [6,7].

It can be observed that the raw sensor data from IoT sensors embed-large scale unclean and useless data. Thus, the raw sensor data need to undergo data cleaning processing, and then data analysis can be performed to obtain relevant information from this cleaned IoT sensor data [8,9]. Further, the large quantity of unwanted and useless data can lead to high computation costs and the overutilization of resources in a constrained IoT sensor network. The most common data processing techniques are data denoising, data imputation, data outlier detection, and data aggregation [10].

It is observed that the raw sensor data exhibit unwanted changes and modifications in the original signal. This raw data signal left untreated leads to expensive resource utilization and computation requirement. As such, the raw sensor signal’s data processing is essential, and a variety of existing solutions are addressed in this paper.

In the IoT sensor network, the nodes are distributed, and several nodes are used to perform the same operation. Hence data integration or fusion from multiple sensors is required to improve accuracy in various IoT-based application services [11,12,13]. For example, in a real-time traffic monitoring system, data patching is useful for data fusion. The previous week’s data are then fused to other data from the time of loss of data. This process involves count-and-classify data, loop-based data, vehicle speed measurement, automated number plate recognition (APNR), etc. As such, in this paper, the details of data fusion are also addressed.

A further dimension of the IoT sensor network addresses the fact that the IoT sensors’ data possess complex properties, such as voluminousness, veracity and velocity. Thus, it is essential to store this data for performing data analysis and achieve the desired outcome for IoT sensor-based applications. IoT sensor network integration with emerging technologies provides efficient methods to handle sensor data’s dynamic and complex nature. Furthermore, machine learning and deep learning techniques provide a promising solution towards the analysis of IoT sensor data [14,15,16]. Incorporating these data analysis techniques results in deep insights into sensor data, and provides good knowledge related to hidden data patterns and further decision-making. In this respect, this paper elaborates on various existing data analysis approaches. 

It is observed that several works exist with each focusing on specific problems and issues associated with IoT sensor data. However, there is a lack of papers offering a complete overview of various IoT sensor data techniques, such as data processing, data analysis and data fusion.

The overall contributions of this paper are listed below:To provide an overview of various data analysis techniques for IoT sensor data;To explain the basic architecture for IoT sensor data processing, fusion, and analysis. Further, the interaction of these modules along with the IoT sensor network;To discuss the IoT sensor output characteristics, such as the voluminous IoT sensor data, heterogeneity, real-time processing, and scalability factors;To explain the mechanism of data processing techniques so as to address various issues in IoT sensor data, such as data denoising, missing value imputation, data outlier detection, and data aggregation;To address the importance of deep learning and machine learning models for IoT sensor data analysis. The need to integrate merging technologies such as cloud, fog, and edge computing towards the efficient computation of data analytical models.

Section 2 elaborates on the basic architecture of IoT sensor data processing, analysis and data fusion. Furthermore, the characteristic of sensor data is discussed in this section. Next, Section 3 details the different data processing methods, such as data denoising, missing data handling, data outlier detection, and data aggregation. Section 4 elaborates on the data fusion methods, and Section 5 presents the data analysis mechanisms. Finally, Section 6 concludes this paper.

## 2. Basic Architecture

In the technology era of the Internet of Things (IoT), the interconnection of physical things with virtual objects aims to enhance human life quality through advanced applications and growing sensor technology, communication networks and processing methodologies. A few examples of such advanced applications include connected smart city, intelligent transport systems, smart healthcare, smart building and smart grids. Towards enabling rapid advancement in IoT, the sensor networks play a vital role by sensing critical data from both the internal functional systems and the external environmental factors [17,18]. The wireless-based sensor network is much more popular, as these networks can be deployed ad hoc without the prerequisite of any infrastructure. The wireless sensor network is capable of self-organizing and can be deployed randomly.

Figure 1 depicts the basic architecture for IoT sensor data processing, fusion and analysis layers. The IoT sensor data layer primarily consists of various IoT sensors that can measure physical surroundings and capture real-time environment changes. The commonly used IoT sensors include temperature, pressure, humidity, level, accelerometer, gas, gyroscopes, motion sensors image, optical sensors, Radiofrequency Identifier (RFID) sensors, and Infra-Red (IR) sensors. The IoT sensors are mainly associated with the microprocessing unit, storage unit, control unit, power system and wireless communication interfaces. The IoT sensor devices are constrained in size, computing power, memory, networking capability and storage space. Wireless communication protocols, such as Wi-Fi, Zig Bee, Bluetooth, Near Frequency Communication (NFC) and LTE/4G mobile technologies, are commonly used for IoT sensor device communication.

The majority of IoT sensor data incorporate real-time processing for industrial applications, healthcare, and scientific activities. For example, the healthcare body sensors to monitor the patients’ critical conditions would generate massively voluminous data. These sensed data must be processed to remove uncertainties for further data analysis, so as to develop knowledge and decision-making. Thus, the data processing layer targets different functions, such as data denoising, data outlier detection, missing data imputation and data aggregation.

The data fusion layer is required to handle various sensor data challenges generated by several heterogeneous sensor devices. The data fusion data aim to integrate true sensor data from heterogeneous IoT sensor devices. The combined data from different sources are then passed to the data analysis layer for efficient knowledge generation and decision-making.

The primary data fusion involves the direct fusion of data sensor data from different sensor devices. It incorporates initial feature extraction which is followed by data fusion. The enhanced method involves feature extraction followed by identity declaration and data fusion [19,20]. This method allows for a high level of inferences of knowledge and much accurate decision-making. In recent years, the adoption of emerging technologies has revolutionized cloud computing, fog computing and edge computing towards IoT sensor data analysis. These enabling technologies provide a pervasive, reliable and convenient platform to handle IoT sensor data’s dynamic, heterogeneous nature [21,22]. As such, the data analytic layer aims at developing smart functionality to address a wide variety of IoT-based applications. The objectives of these platforms are to reduce the computation and storage cost, improve network transmission reliability, reduce the network delay, enhance IoT network security and privacy, ensure scalability, and allow failure- and risk-free IoT solutions.

### Characteristics of IoT Sensor Data

The IoT sensors generate data consecutively or upon the trigger of an external event. The other process involves data generated by sensor nodes that need to be gathered, aggregated, analyzed and visualized to obtain useful information. This information is then interpreted to produce the representable form, that is deliverable, and a reaction towards the external trigger. In addition to the data generated by sensor networks, other sources also have data streams. As such, the data generated are required to be aggregated and warehoused in an unprecedented manner, and streamed at a specific network data rate into remote locations for historical data analysis. However, there are several sensor data characteristics and problems associated with this. The authors in [23] discuss that sensor data exhibits information complexity due to factors like the huge volume, the dynamism of data, real-time updating, critical data aging, and interdependency between different data sources. Generally, the sensors are implanted into the human body, objects or locations. As such, the significant characteristics of IoT sensor data are as given below:

*Technical Constraints*—The limited size of the sensor leads to technical constraints such as computing power, battery power, networking capability, storage capacity and memory. As such, these sensors are highly vulnerable to failure, attacks, and easy breakdown, thus leading to losses of sensor data and inaccurate information;

*Real-Time Processing*—The sensor network will be capable of more complex networking tasks, and can perform the transformation of raw sensor data into more valuable and insightful information in real-time;

*Scalability*—In the physical world, the sensor network includes data sources from numerous sensors and actuators. Sensor networks must be scalable to accommodate the exponential growth of sensors and actuators, data handling, and meet the various objective of IoT-based applications;

*Data Representation*—The general format of sensor data is as a small-sized tuple with structured information. The various representations of sensor data are Boolean, binary, featured values, continuous data, and numeric values;

*Heterogeneity*—IoT sensor data are heterogeneous. There are different data sources, including rigidly structured data sets, real-time data-generating information networks, embedded systems with sensors, social network media data stream, and other participatory sensor networks.

## 3. IoT Sensor Data Processing

In IoT sensor networks, wireless communication protocols are popularly used for the information exchange process. These communication protocols work as unlicensed frequency bands that ease the flexibility and scalability of sensor deployments. However, the utilization of communication protocols for WSN under unlicensed frequency bands causes uncontrollable interference. The interference signals may lead to improper data transmission and sensor data with noise, missing values, outliers and redundancy. This section elaborates on the various data analyses performed to handle IoT sensor data issues such as denoising, missing data imputation, data outlier detection and data aggregation.

### 3.1. Denoising

The voluminous sensor data generated in the IoT network needs data analysis, mostly with real-time decision-making. The characteristics of sensor data are complex, involving high velocities, huge volumes, and dynamic values and types. Further, the sensor data pollute while perpetuating numerous obstacles until producing the required data analysis and real-time decision-making.

Noise is an uncorrelated signal component that enacts unwanted change and modification on the original vectors of the signal. The noise feature leads to the unnecessary processing and utilization of resources for handling the unusable data. The wavelet transform methods are capable of representing the signal and addressing the problem of signal estimation effectively. Significantly, the wavelet transformation preserves the original signal coefficients by removing the noise within the signal. This is achieved by thresholding the coefficient of noise signals, hence the perfect thresholding scheme is essential. The wavelet transformation is a prevalent method to analyze and synthesize continuous-time signal energy. 

Let *e*(*t*), *t* ∊ R represent the signal energy, while it must satisfy the constraint defined as
(1)$${\left||e|\right|}^{2}=\int_{-\infty }^{\infty }{\left|e\left(t\right)\right|}^{2}dt<\infty ,$$
where the signal energy *e*(*t*) that satisfies the constraint in Equation (1) belongs to the squared search space $L^{2}\left(R\right)$. The wavelet transformation is also used to analyze the discrete-time signal energy and eliminate the noise with energy signals. The wavelet transformation method allows us to investigate the signal characteristic by zooming at different time scales. The experimental results exhibited significant improvements in denoising the sensor signals. 

There are two types of wavelet transforms, namely Continuous Wavelet Transform (CWT), which targets signal analysis on a time-frequency level, and Discrete Wavelet Transform (DWT) that targets signal analysis a time level. 

*Continuous Wavelet Transform (CWT)*: In CWT, the signal energy *e*(*t*) is represented using a set of wavelet functions $C=\left\{W_{\psi }\left(\alpha ,\beta \right)\right\},\alpha \in R^{+};\beta \in R$, where α represents the dilation scaling factor, and β represents the shifting time localization factor, while ψ represents the wavelet function. The wavelet coefficient on the time-frequency plane is given by Equation (2).
(2)$$W_{\psi }\left(\alpha ,\beta \right)=\int_{-\infty }^{\infty }\frac{1}{\sqrt{\alpha }}\psi_{0}\left(\frac{\lambda -\beta }{\alpha }\right)e\left(\lambda \right)d\lambda$$
where $\psi_{0}$ represents the shifted and dilated form of the original wavelet $\psi_{0}\left(t\right)$. The CWT is a function controlled by two parameters. The CWT targets to find the coefficients of the original signal *e*(*t*) based on the shifting factor (β) and the dilation factor (α).

*Discrete Wavelet Transformation (DWT)*: The DWT for continuous-time signals refers to transforming signals performed upon a discrete-time signal. The coefficients obtained from this transformation are defined in subset D=Wψ(2α,2αβ),α∈ℤ, β∈ℤ. For a given continuous-time signal e(λ), the coefficients of DWT are obtained using the integration of the subset D, as defined in Equation (3).
(3)w(α,β)=(ψ0(2α,2αβ), e)=∫−∞∞2−α2ψ0′(2−αλ−β)e(λ)dλ
where α indicates the scale factor and β indicates the localization factor. It is to be noted that this involves continuous-time signal e(λ), and not the discrete signal.

The authors in [24] discuss that in some instances the signals received from the IoT sensor devices have a reasonable ratio value of Signal to Noise (SNR), but are unable to achieve the required Bit Error Rate (BER). To overcome such problems, the best solution is to eliminate the inferior wavelet coefficients. This elimination improves the SNR, based on a specific threshold limit. This is possible as the smaller coefficients tend towards more noise data than the desired signal data. Further, it is to be noted that the energy signals are concentrated on a particular part of the signal spectrum. As such, if that specific part of the signal spectrum has been transformed using wavelet coefficients, it improves the SNR value. Further, if the signal function has large regions of irregular noise and small smooth signal regions, then the wavelet coefficients play a vital role in improving the signal energy. Thus, if any signal function is polluted by larger noise, the wavelet coefficients are affected in the more significant part of the wavelet coefficients. The original signal will be contained within the small parts of wavelet coefficients. Thus, maintaining the right threshold limit would eliminate the majority of noise signals and retain the original signal coefficients. In this paper, the authors elaborate on the streaming of sensor data and raw sensor signals to recognize the characteristics and various issues related to noise associated with the sensor signals.

### 3.2. Missing Data Imputation

Imputation is an essential pre-processing task in data analysis for dealing with incomplete data. Various fields and industries like smart cities, healthcare, GPS, smart transportations, etc., use the Internet of Things as a key technology that generates lots of data [25]. The learning algorithms which analyze the IoT data generally assume that the data are complete. While missing data are common in IoT, the data analytics performed on missing or incomplete IoT data may produce inaccurate or unreliable results. Therefore, an estimate of the missing value is necessary for IoT. Three main tasks must be performed to solve this problem. The first step is finding the reason for missing data. Poor network connectivity, faulty sensor systems, environmental factors and synchronization issues are the various reasons for the incomplete results. The missing data are divided into three types: missing completely at random (MCAR), missing at random (MAR), and not missing at random (NMAR). The further step involves studying the pattern of missing data. The two approaches are monotonous missing patterns (MMP) and random missing patterns (AMP). Finally, they form a missing value imputation model for IoT to use the model to approximate the value for the missing data. Within the literature, some missing value imputation algorithms include single imputation algorithms, multivariate imputation algorithms, etc. The traditional imputation algorithm is not suitable for IoT data. There are some algorithms which are mainly used for missing data imputation, and these are given in the next section.

*Gaussian Mixture model:* The Gaussian Mixture Model (GMM) is a clustering algorithm [26]. It is a probabilistic model that uses a soft clustering approach for distributing the data points to different clusters. 

A Gaussian Mixture is defined as follows: G = {*GD*_1_, *GD*_2_, …, *GD*_k_}, where k denotes the number of clusters. Each *GD*_i_ is a Gaussian distribution function that comprises a mean μ, which represents its center, a covariance Σ, and a probability π, which denotes how big or small the Gaussian function will be. Assuming a data set D is generated using GMM with k components, the function f_k_ (*GD*_i_) represents the probability density function of the *k*th component. The probability of *GD*_i_, $P\left(GD_{i}\right)$ generated by GMM, is as given in Equation (4) below.
(4)$$P\left(GD_{i}\right)=\overset{k}{\underset{i=1}{\sum }}\pi_{i}\;f_{i}\;\left(GD_{i}│\mu_{i},\Sigma_{i}\right)$$

To handle the missing data imputation of the IoT sensor data using the GMM model involves five steps, namely instance creation, clustering, classification, distance measuring and data filling. First, the instances in the data set D are divided into two separate data sets, as D_1_ and D_2_. D_1_ contains all the instances without missing values, whereas D_2_ contains all instances in the data set which has the missing values. Secondly, the GMM model based on the EM algorithm is used to cluster the complete data set D1. The cluster center for each cluster is determined. After that, the cluster for each instance in D1 is computed. Third, the incomplete data set D_2_ is taken as a testing set. Each instance in D_2_ is classified according to the clustering result. For instance, $\alpha_{i}$
$\in D_{2,}$
$\alpha_{i}$ belongs to a cluster if it is closer to the cluster center of that cluster by Euclidean distance. In the fourth step, for each instance $\alpha_{i}$ in D_2_, one must determine one or more complete instances that are closest to $\alpha_{i}$ in the same cluster, using Euclidean distance as the distance measure. Finally, one must fill in the missing value of the instance $\alpha_{i}$ by finding the mean value of the closest instance in the cluster.

*Spatial and temporal correlation* [27]: Sensor nodes periodically detect data. Since the sensor data are time-sensitive, different results would be obtained using other sensor data for analysis. The relationship between the sensor nodes in different periods is not the same, so it is necessary to select the correct data to analyze. According to authors, the appropriate sampling of data is required for accurate data analysis. The authors propose the Temporal and Spatial Correlation Algorithm (TSCA) to estimate the missing data. Firstly, it saves all the sensed data simultaneously as a time series and selects the most important series as the analysis sample, which significantly improves the algorithm’s efficiency and accuracy. Secondly, it estimates missing temporal and spatial dimensional values. These two measurements are assigned different weights. Third, there are two strategies for dealing with severe data loss, which boosts the algorithm’s applicability. The basic workflow of the TSCA model is illustrated in Figure 2.

The model as illustrated in Figure 2 assumes all the sensors are in the same communication range. The experiment was conducted on the air quality data set. As a first step, the target data set is extracted from the original data set. A missing data threshold is set, which differs from case to case. In the next step, compute the percentage of missing values. If it exceeds the threshold, then the imputation is ignored; otherwise, the spatial–temporal imputation is carried out. In the next step, the n proximity sensors are computed using the Haversine distance formula. The correlation between the n proximity sensors and the sensor with a missing value is calculated using the Pearson correlation co-efficient. Finally, the complete target data set is constructed and evaluated for accuracy.

The authors in [28] suggested a novel method of nearest neighbor imputation to impute missing values based on the spatial and temporal correlations between sensor nodes. The data structure kd-tree was deployed to boost search time. Based on the percentage of missing values, weighted variances and weighted Euclidean distances were used to create the kd-tree. Figure 3 illustrates the workflow of the spatial–temporal model. The algorithm defined in the proposed model follows the steps, as firstly, it sets the missing value threshold as T. The percentage P of the missing values in the chosen data set is then calculated. If P is within the threshold, then it finds the n proximate sensors through spatial correlation. The correlation between the sensors with the missing values and the n proximity sensors is computed using the Pearson correlation coefficient. The missing sensor data are then imputed by the readings of the proximity sensors corresponding to the time. The output data set is completed. The result is then compared with multiple imputation outcomes. Again, the accuracy is evaluated using Root Mean Square Error (RMSE). 

*Incremental Space-Time Model (ISTM)*: The incremental model discussed in [29] is the model that updates the parameters of the existing model depending on the previous incoming data, rather than constructing a new model from scratch. The model uses incremental multiple linear regression to process the data. When the data arrive, this model is updated after reading the intermediary data matrix rather than accessing all the historical data. If any missing data are found, then the model provides an estimated value based on the nearest sensors’ historical data and observations. The working of the ISTM model has three phases, which are initialization, estimation and updating. In the initialization phase, the model is initialized with historical data. For each sensor p, the historic data and the recording reference are represented as a data matrix. It also computes two intermediary matrices using these data matrices. Next, in the estimation phase, if the sensor’s data contain one missing value, then the model generates an estimated value in real-time. It does this by referring to some data in the database. The estimated value will be then saved in the database. Finally, if the data arriving from the sensor do contain any missing value in the updating phase, then the model updates the new data. It then stores this in a reference database.

*Probabilistic Matrix Factorization*: There are two major advantages of using probabilistic matrix factorization (PMF) [30] for handling missing IoT sensor data. First is the dimensionality reduction, which is the underlying property of matrix factorization. The second is that the original matrix can be replicated using the factored matrices product. This method is used to recover the values missing in the original matrix. PMF is performed on the preliminarily assigned sensors. The neighboring sensors’ data are examined for similarity, and are grouped into a different class of clusters using the K-means algorithm.

The K-means clustering algorithm groups the sensors into separate classes according to their measuring similarity. Analyzing the patterns of neighboring sensors helps to recover missing sensor data. Then, a probabilistic matrix factorization algorithm (PMF) is implemented in each cluster. PMF standardizes the data and restricts the probabilistic distribution of random features. The workflow of the algorithm is illustrated in Figure 4.

In the algorithm, PMF is used to factorize a single matrix into the product of two matrices. The dimensionality can be reduced by factorization. The ability to obtain the original matrix from the product of two factored matrices can be used to recover the missing values in the original matrix. The original matrix is represented as P _NxM_. Now generate two random matrices, U and V, such that P’ = U. V ^T^, where U and V are of dimension NxK and KxM, respectively. K is an integer which represents the number of latent feature column-vectors in U and V. The missing data points in the original matrix are represented as an identity matrix, I having the same dimension as the original matrix P. The values in the I_ij_ matrix are defined using the following rule, as depicted in Equation (5).

(5)$$I_{ij}=\{\begin{array}{c} 1,\;if\;P_{ij\;is\;not\;missing}\\ 0,\;if\;P_{ij\;is\;missing} \end{array}$$

Next, compute the root mean square error (RMSE) between the P and P’, which is given in Equation (6) below.
(6)$$RMSE=\sum_{i=1}^{N}\sum_{j=1}^{M}\;I_{ij}\;{\left(P_{ij}-U_{i}V_{j}^{T}\right)}^{2}$$

The result obtained using Equation (1) is compared with $RMSE_{max},$ which is the maximum acceptable error. The algorithm will terminate if RMSE is ≤
$RMSE_{max}$. Otherwise, the values of U and V are updated using the Equations (7) and (8).
(7)$$U^{\prime }=U_{i}+\gamma .\frac{\partial RMSE_{ij}}{\partial U_{i}}$$
(8)$$V^{\prime }=V_{j}+\gamma .\frac{\partial RMSE_{ij}}{\partial V_{j}}$$
where γ denotes the adjustment factor that decides how much the U and V values need to be adjusted. Steps four to six are repeated until the RMSE is less than or equal to $RMSE_{max}$. A large value of γ may result in low precision, while a value that is too small may result in too many iterations.

The authors in [31] addressed the issue of missing value in IoT sensors. In IoT sensor networks, single-mode failures cause data loss due to the malfunction of several sensors in a network. The authors proposed a multiple segmented imputation approach, in which the data gap was identified and segmented into pieces, and then imputed and reconstructed iteratively with relevant data. The experimental results exhibited significant improvements over the conventional technique of root mean square.

### 3.3. Data Outlier Detection

In the IoT sensor network, the sensor nodes are widely distributed and heterogeneous. It is to be noted that, in a real physical environment, such a setup leads to enormous failure and risks associated with sensor nodes due to several external factors. This causes the original data generated from the IoT sensor network to become prone to modifications and produce data outliers [32]. Therefore, it is essential to identify such data outliers before performing data analysis and decision-making. For this purpose, spatial correlation-based data outlier detection is performed and carried using three popular methods, namely majority voting, classifiers, and principal component analysis.

*Voting Mechanism*: In this method, a sensor node is identified as functioning abnormally by finding the differences in reading with neighboring sensor nodes. According to authors [33], the Poisson distribution is the usual data generation method in various IoT sensor network applications. In the IoT sensor network, the data sets generated consist of outliers for short-term, non-periodic, and insignificant changes in data trends. The simple and efficient statistical methods for outlier detection in the Poisson distribution of the IoT sensor network data set are standard deviation and boxplot. Similarly, in a distributed environment, the Euclidean distance is estimated between the data generated by the faulty sensor node and the data from its immediate neighbor nodes. If the estimated difference in the data exceeds a specific threshold limit, then the data generated by this node are identified as an outlier. Although this technique is simple and less complicated, it is excessively dependent on the neighboring sensor nodes. Furthermore, the accuracy in the case of the sparse network is low.

*Classifiers*: This method involves two steps, firstly training the IoT sensor data using a standard machine learning model. Secondly, the data are detected using the classifier algorithms as either normal or anomaly [34]. The commonly used classifier algorithms is the support vector machine (SVM). Half of the data search space is trained to be standard data. Later, the data are analyzed and classified through SVM for the detection within the trained data of standard data, or otherwise of abnormal data. However, the demerits of the classifier algorithm involve its high complexity in the computation aspect.

*Principal Component Analysis (PCA)*: The objective of PCA is to identify the residual value by extracting the principal components of the given data set. The residual values estimated by the data are evaluated through detection mechanisms such as T^2^ score and squared prediction error (SPE) to identify the data outliers.

In [35], the authors addressed the outlier detection in the IoT sensor data using the Tucker machine and the Genetic Algorithm. Different sensor nodes are involved in the IoT sensor network, exhibiting spatial attributes and sensing data. Furthermore, the sensor data generated are dynamic for the time. Generally, the extensive sensor data contain anomalies due to the mode failure. The conventional means of detecting outliers involves vector-based algorithms. The demerits of vector-based algorithms disturb the original structural information of the sensor data, and exhibit the side effect of dimensionality. The authors proposed a tensor-based solution for the outlier detection using tucker factorization and genetic algorithms. The experimental results demonstrated improvements in the efficiency and accuracy of outlier detection without disturbing the intrinsic sensor data structure.

### 3.4. Data Aggregation

The data aggregation method is referred to as the method that collects and communicates information in a summary form. This can be used for statistical analysis. In IoT, heterogeneous data are collected from various nodes. Sending data separately from each node leads to high energy consumption, and needs a high bandwidth across the network, which reduces the lifetime of the network. Data aggregation techniques prevent these kinds of problems by summarizing data, which reduces the excessive transfer of data, increases the network’s lifetime, and reduces network traffic. Data aggregation in the Internet of Things (IoT) helps to decrease the number of transmissions between objects. This lengthens the lifetime of the network and decreases energy consumption [36]. It also reduces network traffic. 

The data aggregation methods in IoT are classified into the following:(a)Tree-based approach [37,38,39,40,41,42,43]—This approach deploys the nodes in the network in the form of a tree. Here, the hierarchical and intermediate nodes perform the aggregation. The aggregated data are then transferred to the root node. The tree-based approach is suitable for solving energy consumption and lifetime of network problems;(b)Cluster-based approach [44,45,46,47,48]—The entire network is organized as clusters. Each cluster will contain several sensor nodes with one node as the cluster head, which performs the data aggregation. This method aims to carry out the effective implementation of energy aggregation in large networks. This helps to reduce the energy consumption of nodes with limited energy in huge networks. Therefore, it results in a reduced overhead bandwidth due to the transfer of a limited number of packets. In the case of static networks where the cluster structure does not shift for a long time, cluster-based techniques are successful(c)Centralized [49,50]—All sensor nodes in this system send the data to a header node, which is a strong node with all the other nodes. The header node is responsible for aggregating the data and sending out a single packet.

The authors in [51] addressed the issue of data uncertainty in IoT sensor data. Mainly, the data aggregation focused on device to device communication. The proposed technique involves the reconstruction of subspace-based data sampling. Next, the low-rank approximation is performed to identify the dominant subspace. Further, the robust dominant subspace is utilized for reliable sensor data, in an entirely supervised manner. The proposed method exhibits improvements in terms of accuracy and efficiency as regards removing the uncertainties and data aggregation of sensor data from the experimentation results.

## 4. Data Fusion

Data integration or fusion from multiple sensors is required to improve accuracy in various applications. For example, it can be used to trace a target in a military or surveillance system, trace an on-road vehicle’s exact location, find the position of an obstacle in veins of the human body, etc. Various applications of data fusion in different domains are briefly explained as follows [52,53]:The multi-sensor data fusion approach can be applied in ships, aircraft, factories, etc., from a microscopic scale to a distance to hundreds of feet. In these systems, data from electromagnetic signals, acoustic signals, temperature sensors, X-rays, etc., can be integrated for validation and increased accuracy. This integration will increase the accuracy and build trust in the system, which is helpful in the timely maintenance of activities, system fault detection, and remote correction, etc.;Medical diagnosis is a critical system that involves the human body, and is used in disease identification, such as for tumors, lungs or kidney abnormalities, internal diseases, etc., through NMR, chemical or biological sensors, X-rays, IR, etc.;Many satellites, aircraft, and underground or underwater machines use seismic, EM radiation, chemical or biological sensors to collect accurate information or identify natural phenomena in environment monitoring from very long distances;Military and defense services use this technique in ocean surveillance, air-to-air or ground-to-air defense, battlefield intelligence, data acquisition, warning, defense systems, etc., using EM radiation from large distances.

Sensor fusion is the process of combining two or more data sources in a way that generates a more consistent, more accurate, and more dependable estimate of the dynamic system. This estimate gives better results than if the sensors were used individually. The objective of sensor fusion is to minimize cost, device complexity and the number of components involved, and improve sensing precision and confidence.

The data sources can be sensors or mathematical models, and the system state can be acceleration, distance, etc. Four different reasons for using sensor fusion include that (i) it increases the quality of data, (ii) it can increase reliability, (iii) it can measure unmeasured states, and (iv) it can increase the coverage area.

In general, the data fusion methods could be characterized as probabilistic, statistical, knowledge-based, and inference and reasoning methods. The probabilistic methods include Bayesian networks, maximum likelihood estimation methods, inference theory, Kalman filtering, etc. Statistical methods include covariance, cross variance, and other statistical analyses [54]. Knowledge-based methods include artificial neural networks, fuzzy logic, genetic algorithms, etc. [55,56]. Depending on the problem specification, the appropriate data fusion methods are to be chosen. Here, the basic Bayesian and Kalman filter methods are explained.

*Bayesian method*: In multi-sensor data fusion, the essential property of Bayesian statistics is that all unknown variables are considered random variables. The probability distribution function defines what is known about these unknown quantities. For a given probability distribution, the parameters are estimated from the prior distribution, and translates the uncertainty about the parameter values. The basic Bayes law states that
(9)$$P(\alpha |\beta_{1},\beta_{2})\propto P\left(\alpha \right)L\left(\alpha ;\beta_{1}\right)L\left(\alpha ;\beta_{2}\right)$$
where prior density upon α is represented by P(α), the likelihood of α by β is represented by L(α;β), such that likelihood is proportional to P(α|β), and the conditional probability is given by the equation below.
(10)L(α;β)∝P(α|β)

This shows the probability of receiving sensor data β given the a priori value α. The fusion of two different sensor data measurements α and β, given a non-inform prior such as P(α), is then given
(11)$$P(\alpha |\beta_{1},\beta_{2})\propto P\left(\alpha \right)L\left(\alpha ;\beta_{1}\right)L\left(\alpha ;\beta_{2}\right)$$
(12)$$1\times exp\frac{1}{2}{\left(\frac{\alpha -\beta_{1}}{\delta_{1}}\right)}^{2}\times exp\frac{1}{2}{\left(\frac{\alpha -\beta_{2}}{\delta_{2}}\right)}^{2}$$

*Kalman Filter Method*: This is used to estimate the system state when it cannot be measured directly. It is an iterative mathematical process that uses a set of equations and data inputs measured over time, containing noise and inaccuracies. It produces estimates of these unknown parameters that are more accurate than those taken from a single sensor measurement, by using a joint distribution function over each timestep variable. 

It works as a two-step recursive algorithm, with a prediction and update step. In the prediction step, it estimates the current state of the variables along with the noises. Suppose the outcome of the new measurement is observed with some noise or uncertainty. In that case, the estimates are updated using a weighted average, assigning more weight to the estimates with higher certainty and less to estimates with low uncertainty. The estimated uncertainty of the predicted system’s state is represented as a covariance matrix, and the weights are calculated.

The new estimate of the weighted average is obtained from the weighted average that lies between the predicted and measured state, giving better-estimated uncertainty. This process is recursive until the filter follows the model predictions more closely.

The Kalman filter model assumes the evolution of a state at time k from the state at (k − 1), according to the following equation.
(13)$$x_{k}=Fx_{k-1}+Au_{k-1}+w_{k-1}$$
where *F* is the state transition matrix which is applied to previous state $x_{k-1}$ vectors, and A is the control input matrix which is applied to the control vector $u_{k-1}$. $w_{k-1}$ is the noise vector that is assumed to be the multivariate Gaussian distribution drawn from the zero mean with the covariance $Q_{k}$, where $w_{k-1}\sim N\left(0,Q_{k}\right)$. The measurement at time step k is observed as
(14)$$z_{k}=Hx_{k}+v_{k}$$
where H is the measurement matrix, $z_{k}$ is the measurement vector, and ν_k_ is the measurement noise vector that is assumed to be a multivariate Gaussian distribution drawn from the zero mean with the covariance R, i.e., $v_{k}\sim N\left(0,R\right)$.

The objective of the Kalman filter is to provide an estimate of $x_{k}$ at time k, given the initial estimate of $x_{0}$, the series of measurements $z_{1}$, $z_{2}$, …, $z_{k}$ and the information of the system described by F, A, H, Q and R.

Three basic approaches can be used in multi-sensory data fusion or integration. 

*Direct fusion*: In this approach, all sensors are associated among themselves for classification. Thereafter, data-level integration or fusion happens. Once the associated data sensor’s data are integrated, then data features are extracted. These data features help in the identification of objects with sensors. This direct fusion process is also known as joint identity declaration, where multiple sensors’ association identities are declared jointly. The direct fusion process is formally designed as shown in Equations (15)–(18).
(15)$$O:\;S_{i}\to S_{j}\;\forall i\neq j\;i\in \left\{1,2,3,\ldots ,n\right\}\;and\;j\in \left\{1,2,3,\ldots ,n\right\}$$
(16)$$P:\;f_{fe}\;\left(f_{df}\left(f_{A}\left(O\right)\right)\right)$$
(17)$$ID_{data\_extraction}\left(S_{i}\right)=g\left(P\right)$$
(18)$$Q:JID_{declaration}\left(S_{i}\right)$$
Here, $S_{i}$ represents the *i*th sensor considered in the data fusion process, and *O* is the outcome of the *i*th sensor from *j*th sensor mapping. $f_{A}\left(.\right)$, $f_{df}\left(.\right)\;and\;f_{fe}\left(.\right)$ are the data association, internal data-level fusion, and feature extraction functions in the complete data fusion process. Individual sensor data identification is achieved by applying the identity declaration function (g) over the feature extraction outcome (P). Finally, *Q* is the outcome of joint identification in the direct fusion approach, using the function $JID_{declaration}$(.) and outcome Q.

*Feature extraction followed by fusion*: In this approach, the sensor’s data features are extracted initially, followed by feature-based data association. These associations facilitate the fusion of the data based on the feature’s association and identity declaration. Finally, feature-based data fusion and identity declaration result in the sensor’s joint identify declaration and data integration. This approach is formally presented as shown in Equations (19)–(22).
(19)$$R:\;f_{fe}(S_{i})\;\forall \;i\in \left\{1,2,3,\ldots ,n\right\}$$(20)$$U:g(f_{A}(R))$$(21)$$V:H\left(f_{A}\left(R\right)\right))$$(22)Q:I(U,V)
Here, *R* is the outcome of feature extraction from each sensor. H(.) is a function to integrate feature-level data fusion. Finally, the data fusion outcome *Q* for the joint identification declaration is computed by applying a parallel integration function I(.) over the feature-level fusion outcome (*U*) and identity declaration outcome (V).*Feature extraction followed by identity declaration and fusion for high-level inferences or decisions*: In this approach, the sensor’s data features are extracted initially. After that, individual sensor data identities are declared from extracted features. These unique identities help in data association or in searching the required data. A unique identity-based data association is used for declaration-level fusion and identity declaration. Both of these processes result in a joint identity declaration. This approach is formally presented as shown in Equations (23)–(28).
(23)$$R:\;f_{fe}(S_{i})\;\forall \;i\in \left\{1,2,3,\ldots ,n\right\}$$(24)U:g(R)(25)$$S_{i}^{ID}=J\left(U\right)$$(26)$$V:H\left(f_{A}\left(U,\;S_{i}^{ID}\right)\right)$$(27)$$W:g\left(f_{A}\left(U,\;S_{i}^{ID}\right)\right)$$(28)Q:I(W,V)
The multi-sensor data fusion approach works in a multi-layered architecture as well. In multi-layered architectures [57], multi-sensory data fusion or integration approaches work at different layers. For example, the multi-layered components of the data fusion architecture, proposed by Joint Directors of Laboratories (JDL) [58], are briefly explained as follows.*Object Refinement*: This is considered as layer-1 processing in the JDL process model. Here, sensor-level data are combined to achieve better reliability, accuracy, position estimation, velocity measurement, attribute evaluations and identity exploration.*Situation Refinement*: This is a layer-2 concept in the JDL process model. According to this layer, a dynamic process is applied to describe the present relationships among various entities considered in the experimentation.

A list of associated events in the context of the environment has been prepared as well, as follows.

*Threat Refinement*: This is a layer-3 concept in the JDL process model. In this layer, present situations are explored in such a way that future trends can be observed. For example, the current military and defense deployment can give an idea about what the implications will be of enemy threats. Who will become a friend or foe? What opportunities will arise if operations are planned soon or over a long time? etc.*Process Refinement*: This is a layer-3 concept in the JDL process model. In this layer, a meta-process is executed. In this meta-process, the overall data fusion processes for assessing and improving the real-time system or sub-system performances are performed.

The layering process used for data fusion in the JDL architecture, the source-preprocessing, database management systems, and human–computer interaction (HCI) are essential components. Source-preprocessing manages the resources and related data for any redundancy, duplicity or loss. Database management systems store the required and necessary information. HCI involves humans in controlling, monitoring and maintaining the systems.

## 5. Data Analysis

According to the authors in [59], the IoT sensor networks’ key problems are the scalability and accuracy of sensor data. These challenges are addressed through sensor data mining and analysis techniques, such as data gathering, cleaning issues, data management, knowledge discovery, and data mining. In this aspect, machine learning and deep learning models play a vital role in obtaining the results, which include knowledge generation and decision-making. 

*Machine learning models*: The authors in [60] addressed the issue that the IoT sensor data require an efficient mechanism for deriving meaningful information from data. In the case of IoT sensor data analytics, the machine learning models are required to be executed within the sensor embedded processor. This requires the customization of system programs and an efficient data structure to handle the features of real-time IoT sensor data. As such, the authors proposed the Gaussian Mixture Model (GMM) as a solution to handle the various sensor data features. Furthermore, the real-time cooperation of hardware and software, and continuous machine learning algorithms, were carried out for data training and the classification of the resultant data for IoT-based applications. 

*Deep Learning Models*: The authors in [61] proposed feature learning for IoT sensor data using the deep learning mechanism. The IoT sensor data provide unclear features that need to be made accurate through real-time deep learning-based classification. However, the deep learning models are computationally expensive in terms of execution within resource-constrained sensor boards. As such, the authors proposed a preprocessing mechanism in the spectral domain, and then deep learning models were executed. 

*Neural Network for IoT Sensor Data Processing/Analysis*: Artificial neural networks are well suited for function approximation and pattern recognition problems using supervised learning techniques. The architecture of ANN includes an input layer, an output layer, and one or more hidden layers. Convolutional neural networks (CNN) consist of a network layer that is used for convolution operation applied to two-dimensional or one-dimensional sensor data [62]. The convolution operator is used to learn the local patterns from the data. Depending upon the available data, the right variant of CNN is used. CNN is mostly suited to the image data used for the analysis of image sensor data analysis. Most of the IoT devices, such as drones, smart cars, smartphones, etc., are equipped with cameras. Many IoT applications, like floods or landslides, forest fire prediction through drone images and traffic management, use vehicle cameras. CNN takes an image/speech/signal which is 2D or 1D, and the high-level features are extracted through a series of hidden layers. The hidden layers include the convolution layer and a fully connected pooling layer at the end. The convolution layer has a set of filters, known as a learnable parameter, which is the core of the CNN that filters the high-dimensional data to a lower dimension, which helps extract the most appropriate feature from the input image.

The recurrent neural network (RNN) is a neural network which learns from sequences or time series data [63]. Some tasks, such as detecting the driver’s behavior in smart vehicles, identifying some movement patterns, or estimating the electricity consumption of the household environment, may depend on previous samples for prediction. In these cases, RNN would be the right choice. It can handle sequences of variable length. RNNs are well-suited for predicting the time series sensor data. The input to an RNN consists of the present sample as well as the sample described previously. In other words, the output of an RNN at step t−1 in time influences the output at step t in time. Each neuron has a feedback loop, which returns the current output as an input for the next step. This structure can be expressed so that each neuron in an RNN has an internal memory that preserves the computational information from the previous input. Long short-term memory (LSTM) is a variant of RNN that can look back for a long time before predicting. LSTM can efficiently retrieve and identify features for the prediction from the sensor data.

Autoencoders (AE) is a neural network that consists of the same number of input and output layers connected through a series of hidden layers [64]. The autoencoders can construct an input at the output layer. Due to this feature, the autoencoders are used in industrial IoT for many applications, such as fault detection in hardware devices, anomaly detection in the performance of assembly lines, etc. Generative adversarial networks (GANs) contain two neural networks, namely the generative and discriminative networks, which mainly aim to produce synthetic and high-quality data [7]. They work on the principle of min–max games, where one network tries to maximize the value function, and the other network wants to minimize it.

Applying GAN for IoT applications involves generating entirely different data based on the available original data set. The general applications of GAN include localization formulation and wayfinding in graph-based networks [65,66]. In these applications, the generator network module of GAN estimates all the possible feasible paths that exist between two nodes. The discriminator module of GAN performs optimal path identification between the source and the target node. GANs are widely used in assistant systems for disabled persons. Further, the authors discuss GAN’s implementation in converting the image to audio applications for visual impaired humans. Another example is the generation of descriptive information in text form out of a given original image. The authors applied GAN for image processing applications, where the objective is to identify the forged celebrity consisting of different pictures of the authentic celebrity.

The IoT sensor network-based applications involve dynamic factors, distributed services, and real-time responsive mechanisms [67]. Hence, there is a middleware layer requirement between IoT-based applications and the underlying IoT sensor data. Further, the scalability addresses huge volumes of data that are obtained from the IoT sensor network. To tackle dynamic, real-time processing and scalability issues, solutions through the integration of IoT sensor networks with other emerging technologies, such as cloud computing, fog computing and edge-based data analysis, are required. 

*Cloud-Based Data Analysis*: The authors in [68,69] proposed cloud-based data analytics for the Internet of Things. To perform efficient data analysis requires a hierarchical and distributed model for the data processing systems. This method is replicated in several virtual machines that re asocial with a remote cloud data center. Then, without any prior data knowledge, the cloud-based analytics model can handle dynamic data processing and scalability issues. The cluster-based operations involve breaking more complex mathematical processes into simpler small tasks, and executing them in different clusters, thus reducing computation cost.

Further, to handle the huge volume of data generated by the IoT sensors, efficient solutions based on cloud computing necessitate the application of big data and massively parallel distributed system technologies. The authors in [70] address the storage of massive quantities of high-velocity and complex sensing data generated by IoT sensor systems. These papers presented data fusion through efficient curation techniques that integrate IoT and the huge sensor data on the remote cloud server, which thus enables the system to provide efficient services for IoT applications. Various issues, such as sensor network scalability, data cleaning, data compression, data storage, data analysis and data visualization, were addressed through the data curation of IoT sensor data into the cloud [71]. The authors in [72] addressed the data acquisition of resource-constrained, distributed IoT sensors.

Further, the possibility of in situ data analysis was discussed to overcome the network congestion problem, with context-aware and real-time processing, in the context of fog and edge computing. Here, surveillance camera sensors were considered for achieving end-to-end optimization. Multiple feature extraction techniques and various classification algorithms were considered, as were the proposed processing depth and amplification of gain through efficient methods. The experimentation result showed a 4.3-fold reduction in power consumption compared with other designs. 

It is to be noted that the IoT sensor data consists of complex time series data, and thus requires efficient data analysis mechanisms. The authors in [73,74] presented a data analytic framework that involved a multi-dimensional feature handling, selection and extraction model. The papers also discussed the dynamic data analytic model for IoT sensor data prediction. The IoT sensor data are represented in time series, and involve complex data analysis mechanisms. The authors proposed a prediction model named the Adaptative Sliding window algorithm, and applied it to the sensor data for feature selection. The experiment was conducted on sensor data obtained from the Intel Berkley Research lab, and results exhibited more than 98% accuracy. The second experiment conducted on Chicago Park Weather Data exhibited more than 92% accuracy. The authors in [75,76] aimed to reduce the data transfer and processing in a remote cloud by efficient fog-based data analytics for IoT sensor data. The paper argued that uploading sensor data into a remote cloud over the Internet adds no value due to the underlying network complexity.

Further, the authors proposed homoscedasticity detection techniques for multi-dimension feature extraction in IoT sensor data. This method, along with neural classifiers, promised to be efficient in extracting important sensor data events in real-time processing. The authors in [77] discussed the pervasive nature of IoT sensor networks and generated a huge volume of sensor data. These sensor data, however, exhibited data redundancy and data outliers that largely degraded the overall performance of the IoT sensor networks. As such, the authors proposed a data analysis framework that aimed at recursively updating and adapting to the dynamic changes in the IoT system’s surroundings. In this, the authors proposed data aggregation through principal component analysis, which involved principal component extraction. Further, the squared prediction error score was proposed in order to identify the data outliers. In real-time experimentation, the proposed data analytics framework exhibited efficient data aggregation and data outlier detection with high accuracy.

*Fog-Based Data Analysis*: The authors in [78,79,80] presented fog-level-based IoT sensor data processing. Here, the senor data features are extracted and processed to classify different signal patterns using a neural network. Consequently, based on the output of neural network classification, event identification and decision-making are performed at the fog level. These sensor signals are classified as normal or abnormal, and accordingly, the alert event is initiated by the fog nodes. The few example applications include smart home systems, fire alarm systems, surveillance systems and air monitoring systems. The objective of fog-based IoT sensor data analysis is to carry out real-time data analytics at the fog device-level, and decision-making as to the alerts used. The authors in [81,82] introduce a content orchestration mechanism that estimates an opportunistic fog node to offload data. The mechanism considers the available bandwidth delay, the cost and the quality of service metrics to determine the opportunistic fog node. Further, the decision-making module enables periodic reports from nodes that serve as input values for a real-time analytical process, and computes the factors for improving the quality of service.

*Edge-Based Data Analysis*: The authors in [83,84,85] series prediction, forecasting, and event-based data handling. For this purpose, many solutions, such as stochastic and regression models, can be performed at the edge level of the IoT sensor network. In this, the data analysis is carried out at the low-level sensor nodes. In the case of time series data forecasting, the two specific prediction models are carried out simultaneously in the sensor devices and the base station nodes. Based on the difference between the values predicted by some constant factor, the base station will be updated with new data. This ensures a low computation cost and resource utilization in an inefficient manner. The authors of [86] present trends in IoT-based solutions from the perspective of health-care. This research aims to provide efficient services within an Internet of Medical Things application, utilizing the edge computing paradigm. In this paper, the authors proposed a three tier IoT architecture that included edge devices, fog network and cloud servers. The sensors act as edge devices and are connected to fog networks for real time data analysis. For high-performance computation and increase storage capacity, these edge devices are connected to the cloud remote servers. The authors present an energy-efficient IoT data compression approach for edge-based machine learning. This research aims to reduce the cost of data transmission, therefore increasing the overall efficiency of data offloading in a cloud-based IoT network. The study applies a fast error-bounded lossy compressor on the collected data before transmission. Later, they apply supervised machine learning techniques to re-build the data. Their experimentation shows that the amount of data traffic is significantly reduced by a factor of 103.

*Semantic Analysis of IoT Sensor Data*: The increasing amount of sensory data arises from making data and applications readily accessible and understandable to future users [87,88]. The semantic enhancements structure and organize the data. This also allows interoperability between machines. The benefit of applying semantic technology to sensor data is the conceptualization and abstract interpretation of the raw data, making them computer-definable, and interlinking the data with existing data web resources. The role of semantics offers new methods for information processing and exploration. It also turns information into actionable knowledge. The most common approaches include (i) linked open data, (ii) real-time and linked stream processing, (iii) rule-based reasoning, (iv) machine learning-based approaches and (v) semantic-based reasoning. The semantic analysis method includes the IoT resources, services, and related processes described using semantic annotations. To add sense to the IoT data, various tools, including observation and measurement data, need to be connected. By applying significant inference and analysis mechanisms to the IoT data, one can also achieve data processing for different domains. This also allows access to domain information and related semantically enriched representations for other entities and/or existing data (on the web). Linked data is a process that connects to different resources and is currently being implemented on the Internet. The linked data approach allows for the interconnection of resources defined by other models and ontologies. The provision of automated resource tagging mechanisms using the principles available, such as linked data and the specification of automated mechanisms of association between various resources (e.g., place, style, provider, and other common properties), render IoT data accessible across different domains. Processing and evaluating semantic descriptions for information extraction, and enabling enhanced interactions with IoT resources, depends on the efficient querying, analyzing and processing of semantic data and resource linkages. Here, the primary objective is to implement the resource-constrained middleware to allow the semantic IoT linked data analytics. This dynamically injects semantics to enrich and make sense of the IoT raw sensor data. The Semantic Sensor Network (SSN) ontology is used as the gateway to represent the sensor’s properties and observations.

## 6. Conclusions

The IoT sensor network’s paradigm shift towards emerging technologies, such as cloud, fog and edge computing, leads to elevated sophistication in data processing, data fusion and sensor data analytics. This paper presents deep insight into the necessity of these processes. The basic architecture of IoT sensor data processing, fusion and analysis has been explained to elaborate on each of these processes’ functionalities. Next, the characteristics of IoT sensor data, such as the voluminous size of the IoT sensor data, heterogeneity, real-time processing and scalability, were explained. The paper provides an overview of various data processing techniques of IoT sensor data, such as data denoising, missing data imputation, data outlier detection and data aggregation. In addition to the data processing techniques, the definitions and processes of data fusion in IoT sensor-based networks have been presented. Further, the paper elaborates the need for enhancing IoT sensor data analysis through emerging technologies, such as cloud, fog and edge computing. Next, this paper presented a novel case study on IoT sensor-based drone networks for centralized traffic monitoring, control and data analytics. The future scope of this work includes addressing the security and privacy challenges of IoT sensor data. Further, there is a need for distributed blockchain technologies for the failure- and risk-free computation of IoT sensor data, which can be addressed as an extension to this present work.

## Figures and Tables

**Figure 1 sensors-20-06076-f001:**
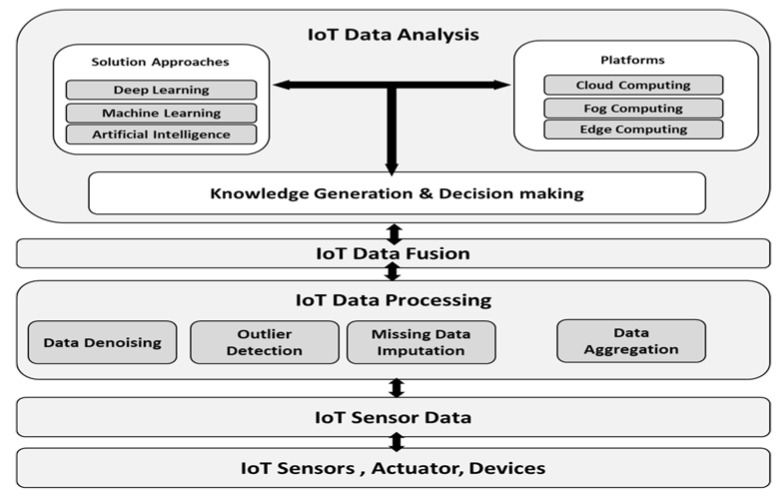
The basic architecture for IoT sensor data processing, data fusion and data analysis.

**Figure 2 sensors-20-06076-f002:**
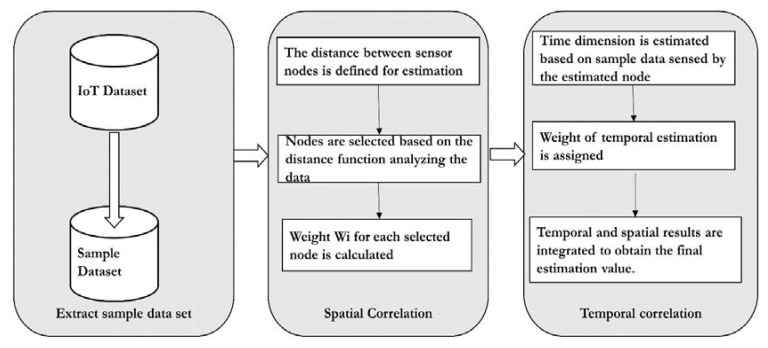
The workflow of the Temporal and Spatial Correlation Algorithm.

**Figure 3 sensors-20-06076-f003:**
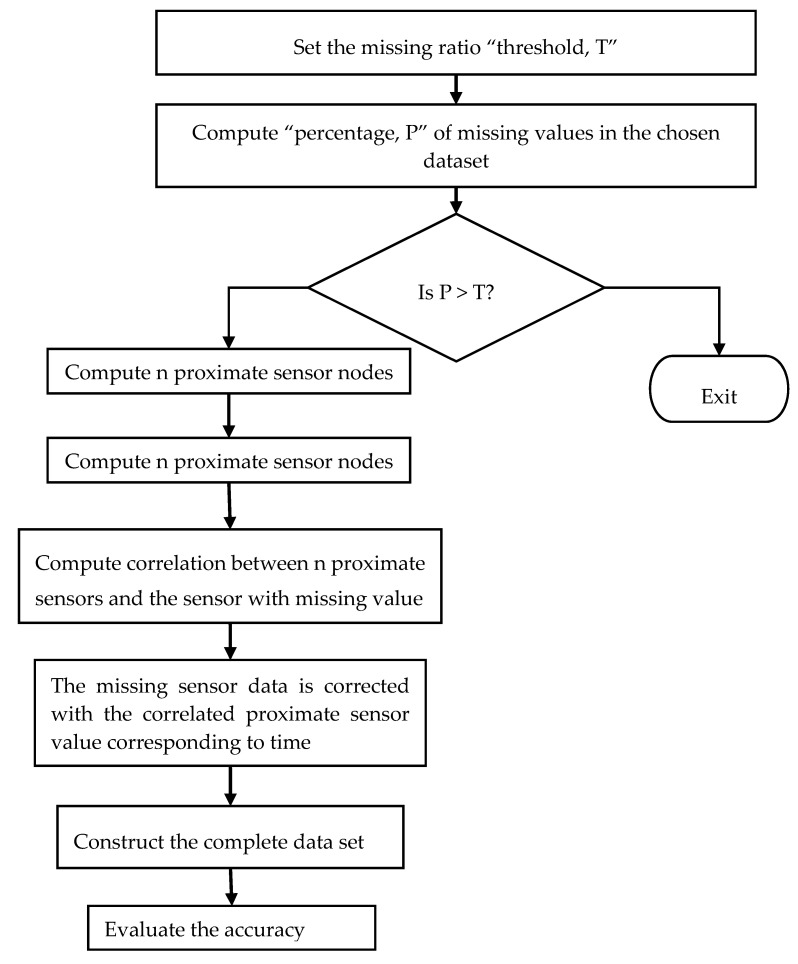
The workflow of the spatial–temporal model.

**Figure 4 sensors-20-06076-f004:**
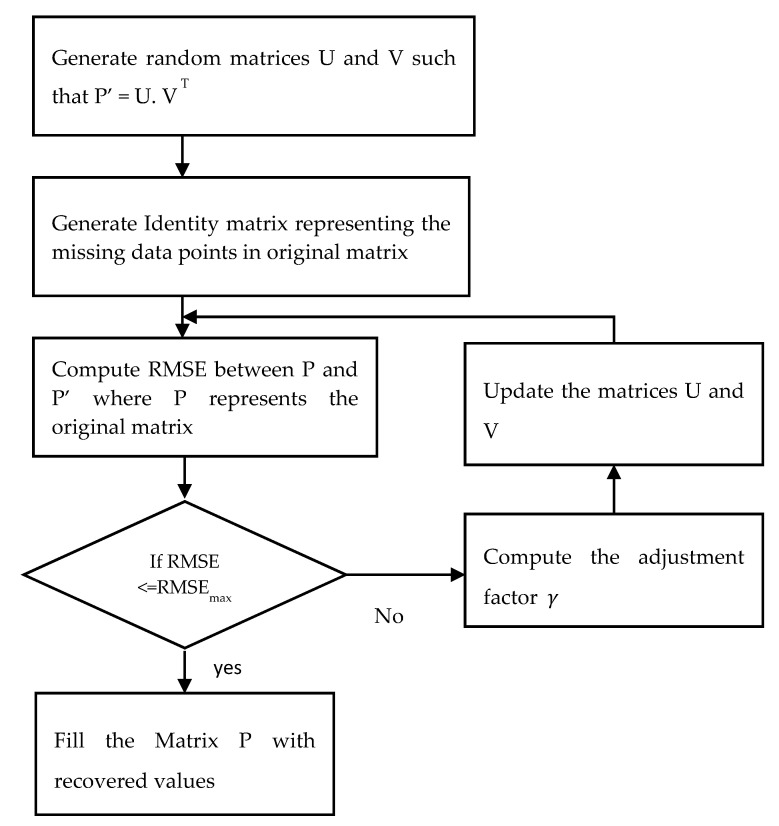
The workflow of the PMF model.

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
