# Peer review of "An Overview of IoT Sensor Data Processing, Fusion, and Analysis Techniques"

_sensors, 2020, doi:10.3390/s20216076_

Round 1

Reviewer 1 Report

The authors give a survey of data processing, fusion and analysis techniques in IoT sensor fields, propose a five-layer framework for IoT sensor road traffic management application and its simulation model. However, I think it needs major revision due to its limited contribution and weak evaluation.

The paper's contribution is limited. For instance, various characteristics of IoT sensor data have been discussed or summarized in existing IoT survey papers, thus the 2nd contribution in the paper is not reasonable. Even the description of size characteristic in lines 157~160 is irrelevant to the subject.

Some claims are inconvincible or vague and the authors do not explain why they make these decisions. For example, the authors claim the other approaches in the table 1 cannot be considered for real-time decision making, but the given reason seems inconvincible. Enough proofs should be provided. Moreover, what do the paragraphs in Lines 560~584 relate to its context? For another example, Figure 3 is questionable and lack of description, thus difficult for understanding.

The evaluation is really weak, the authors should compare the proposed method with other ones, such as the approaches listed in table 1. Moreover, the authors claim that they compared packet transmission delay with the existing work [44], which did not measure the packet transmission delay. In addition, the authors mistook the authors of the reference [44].

Furthermore, the presentation needs to be improved. For instance,
Line 177: The increased interference in sensor network further lead to ...
Lines 280~281: The [28] sample data used for the analysis needs to determined. In proposes ...
Line 765: SSN ontology...
Line 775: the application like loop detector and global positions systems (GPS) signal processing invovle isolated independent experimentations...

Author Response

The authors give a survey of data processing, fusion and analysis techniques in IoT sensor fields, propose a five-layer framework for IoT sensor road traffic management application and its simulation model.

Response: Thank you for the valuable review comments.

-----------------------------------------------------------------------------------------

Comment#1.1:

However, I think it needs major revision due to its limited contribution and weak evaluation.

Response#1.1:

We have updated the list of contributions and enhanced the evaluation approach in the introduction section. The changes are highlighted in yellow colour.

-----------------------------------------------------------------------------------------

Comment#1.2:

The paper's contribution is limited. For instance, various characteristics of IoT sensor data have been discussed or summarized in existing IoT survey papers, thus the 2nd contribution in the paper is not reasonable.

Response#1.2:

We have updated the list of contributions and enhanced the evaluation approach in the introduction section. The changes are highlighted in yellow colour.

-----------------------------------------------------------------------------------------

Comment#1.3:

Even the description of size characteristic in lines 157~160 is irrelevant to the subject.

Response#1.3: Thank you for highlighting the error. We made changes in subheading as “technical Constraint” in place of “size”.

-----------------------------------------------------------------------------------------

Comment#1.4:

Some claims are inconvincible or vague and the authors do not explain why they make these decisions. For example, the authors claim the other approaches in the table 1 cannot be considered for real-time decision making, but the given reason seems inconvincible. Enough proofs should be provided.

Response#1.4:

Many thanks for your comments.

Table 1 approaches are not found to be suitable for real-time decision making in drone-based traffic system. Most of these approaches are used for object detection, data collection and analysis. There is need of some mechanism to automate the control of traffic through drones.

Necessary modifications are made in the revised article. Kindly see section VI.

-----------------------------------------------------------------------------------------

Comment#1.5:

 Moreover, what do the paragraphs in Lines 560~584 relate to its context?

Response#1.5:

Many thanks for your comment.

The necessary explanations are added in section IV

-----------------------------------------------------------------------------------------

Comment#1.6:

For another example, Figure 3 is questionable and lack of description, thus difficult for understanding.

Response#1.6:

The figure 3 is corrected. From line number 306-317, the explanation is added for the figure 3 and highlighted.

-----------------------------------------------------------------------------------------

Comment#1.7: The evaluation is really weak, the authors should compare the proposed method with other ones, such as the approaches listed in table 1. Moreover, the authors claim that they compared packet transmission delay with the existing work [44], which did not measure the packet transmission delay. In addition, the authors mistook the authors of the reference [44].

Response#1.7:

Many thanks for your suggestion.

Evaluation is revised and compared with other method. Yes, It is taken from table 1 only. The necessary modification and comparative results analysis is done in revised article.

-----------------------------------------------------------------------------------------

Comment#1.8:

Furthermore, the presentation needs to be improved. For instance,

Line 177: The increased interference in sensor network further lead to ...

Lines 280~281: The [28] sample data used for the analysis needs to determined. In proposes ...

Response#1.8:

The sentences are rewritten with appropriate grammar and are highlighted.

------------------------------------------------------------------------------------------------------------

Comment#1.9:

Line 765: SSN ontology...

Response#1.9:

Many thanks for your suggestion.

The necessary change has been made in the revised article.

-----------------------------------------------------------------------------------------Comment#1.10:

Line 775: the application like loop detector and global positions systems (GPS) signal processing involve isolated independent experimentations...

Response#1.10:

Many thanks for your suggestion.

The presentation has been improved in the revised article. Kindly see section VI.

-----------------------------------------------------------------------------------------

Reviewer 2 Report

In this papers the authors present and extensive overview of IoT sensor techniques for the management and processing of data, with an example case study in road traffic control problem.

Unfortunately, the paper requires an extensive English language and style editing to make it readable and understandable. Many parts are difficult to read due to this problem. It also makes difficult to assess what the contributions of this paper really are.

For the most part, the paper feels like a Review-type paper, not as a Research article. The authors must edit their paper to make the contribution clearer. What does the Case study provide that it is exactly new? Are there new insights, new knowledge? Or it is just a case study that demonstrates what the overview section describes?

If the paper is intended to be a Research paper, it is suggested that authors shorten the overview section, and limit it to the previous works and concepts related to their main contribution, therefore it could be clear to the reader in what specific areas the authors are making a contribution to the state-of-art.

Author Response

Comment#2.1:

Unfortunately, the paper requires an extensive English language and style editing to make it readable and understandable. Many parts are difficult to read due to this problem. It also makes difficult to assess what the contributions of this paper really are.

Response#2.1:

Thanks for your valuable comments. The English language and grammar are rectified appropriately.

---------------------------------------------------------------------------------------------------

Comment#2.2:

For the most part, the paper feels like a Review-type paper, not as a Research article. The authors must edit their paper to make the contribution clearer.

Response#2.2:

Thank you for pointing out. The authors agree to the comment. The paper is of type “Review”.

---------------------------------------------------------------------------------------------------

Comment#2.3:

What does the Case study provide that it is exactly new? Are there new insights, new knowledge? Or it is just a case study that demonstrates what the overview section describes?

Response#2.3:

Many thanks for your comment.

The purpose and goal of case study is mentioned in the revised article. Kindly see section VI.

---------------------------------------------------------------------------------------------------

Comment#2.4:

If the paper is intended to be a Research paper, it is suggested that authors shorten the overview section, and limit it to the previous works and concepts related to their main contribution, therefore it could be clear to the reader in what specific areas the authors are making a contribution to the state-of-art.

Response#2.4: This paper is of survey type with sample case on road traffic monitoring and control system.

-------------------------------------------------------------------------------------------

Reviewer 3 Report

In Figure 1, why did you make seperated modules for Deep Learning, Machine Learning, and Artificial Intelligence. It is not familiar to me. I think that DL is a subset of ML, and ML is a subset of AI. It can make a confusion to reader.

In Figure 3, there are some texts are missing. Please check the figure.

Author Response

Comment#4.1

In Figure 1, why did you make separated modules for Deep Learning, Machine Learning, and Artificial Intelligence? It is not familiar to me. Think that DL is a subset of ML, and ML is a subset of AI. It can make a confusion to reader.

Response#4.1

We agree with reviewer that DL is a subset of ML, and ML is a subset of AI. However, here our intension, is to represent different algorithms under each model. That is, ML means classification, regression, clustering methods. DL algorithms such as LSTM, CNN, RNN model. AI algorithms such as NN, fuzzy logic etc.

---------------------------------------------------------------------------------------------------

Comment#4.2

In Figure 3, there are some texts are missing. Please check the figure.

Response#4.2

The contents of the figure #3 are rectified for clarification.

Reviewer 4 Report

General comments:

The basic architecture is well described.

However, in my opinion, the results of the application of the proposed approach, and therefore also the conclusions, are not adequately supported by an adequate experimental study.

The case study analyzed by the authors does not use a real data sample, but a hypothetical road network, of which no information is provided (i.e. network length, number of links, number of intersections, traffic volumes, etc.)

The paper is focused on an overview of IoT sensor data processing, fusion, and analysis techniques.

Despite a five-layered framework being presented based on IoT-Sensor network for centralized traffic monitoring and controlling, it is not clear how this is applicable to a real context.

In order to better evaluate the proposed approach, I suggest exploring the five-layered framework by better describing the individual phases and applying them to a more detailed study context. The simulation of the system described in the paper does not clarify some important aspects of the approach (i.e. data collection and data analytics).

The paper is well written and structured; however, it needs minor changes.

Please, revise the English style.

Author Response

Reviewer#4

Comment#3.1

The basic architecture is well described.

Response#3.1

Thank you, for your encouraging comment.

---------------------------------------------------------------------------------------------------

Comment#3.2

However, in my opinion, the results of the application of the proposed approach, and therefore also the conclusions, are not adequately supported by an adequate experimental study.

Response#3.2

Thank you for the review comment. The changes are made appropriately in Case study experimentation section (VI) and conclusion section (VII).

---------------------------------------------------------------------------------------------------

Comment#3.3

The case study analysed by the authors does not use a real data sample, but a hypothetical road network, of which no information is provided (i.e. network length, number of links, number of intersections, traffic volumes, etc.)

Response#3.3

Many thanks for your suggestion. The necessary modifications are done in the revised article. The recommended parameters are specified as well.

---------------------------------------------------------------------------------------------------

Comment#3.4 [KRL]

The paper is focused on an overview of IoT sensor data processing, fusion, and analysis techniques.

Despite a five-layered framework being presented based on IoT-Sensor network for centralized traffic monitoring and controlling, it is not clear how this is applicable to a real context.

Response#3.4

Many thanks for the comments. The section (I), section (VI) and Section (VII) are rectified as per the suggestion of the reviewer.

---------------------------------------------------------------------------------------------------

Comment#3.5

In order to better evaluate the proposed approach, I suggest exploring the five-layered framework by better describing the individual phases and applying them to a more detailed study context. The simulation of the system described in the paper does not clarify some important aspects of the approach (i.e. data collection and data analytics).

Response#3.5

The data collection is performed using AnyLogic simulation (specified in revised article as well). Data analytics part is modified for better explanation in revised article.

-----

Round 2

Reviewer 1 Report

The paper seems more like a review paper than a research article because the contribution in related work is more than the value of the case study. In fact, the case study is weak in novelty and evaluation.

It is suggested that authors modify or delete the case study section for a review paper, or for a research paper, they focus on and make clear their main contribution compare to the state-of-the-art.

Author Response

Comment #1.1: The paper seems more like a review paper than a research article because the contribution in related work is more than the value of the case study. In fact, the case study is weak in novelty and evaluation.

It is suggested that authors modify or delete the case study section for a review paper, or for a research paper, they focus on and make clear their main contribution compare to the state-of-the-art.

Response #1.1:  Thanks for your valuable time for the review of the paper. Yes, you are right that the paper is review paper and our target via this paper is to present one stop comprehensive guidance to researchers, practitioners to take up IoT baseline and strong technical knowledge for undertaking research and that’s why we have presented to the best possible manner the latest related works (UPDATED MORE) in the revised paper with regard to Sensor Data Processing, Fusion and Analysis.

And with regard to your suggestion, we have removed the CASE STUDY and now the focus is more clear and more directed towards the central theme of the paper undertaking the related terminologies to the very best of our knowledge.

Reviewer 2 Report

Thanks for clarifying that this is a Review type of paper. However, the existence of a review paper should be justified, when compared with similar previous publications.

Are there previous review works on the same topic? Is this review its first of its kind? Does the proposed review paper offers a more complete, or updated analysis, or with a different emphasis from previous reviews? Some comments could be incorportated in the introduction in order to highlight better the contributions in terms of review content.

Authors could consider the following references if they offer similar, overlapping themes:

Djedouboum, A.C.; Abba Ari, A.A.; Gueroui, A.M.; Mohamadou, A.; Aliouat, Z. Big Data Collection in Large-Scale Wireless Sensor Networks. Sensors 201818, 4474.

Abu-Elkheir, M.; Hayajneh, M.; Ali, N.A. Data Management for the Internet of Things: Design Primitives and Solution. Sensors 201313, 15582-15612.

H. Cai, B. Xu, L. Jiang and A. V. Vasilakos, "IoT-Based Big Data Storage Systems in Cloud Computing: Perspectives and Challenges," in IEEE Internet of Things Journal, vol. 4, no. 1, pp. 75-87, Feb. 2017, doi: 10.1109/JIOT.2016.2619369.

M. Ma, P. Wang and C. Chu, "Data Management for Internet of Things: Challenges, Approaches and Opportunities," 2013 IEEE International Conference on Green Computing and Communications and IEEE Internet of Things and IEEE Cyber, Physical and Social Computing, Beijing, 2013, pp. 1144-1151, doi: 10.1109/GreenCom-iThings-CPSCom.2013.199.

Author Response

Comment# 2.1: Thanks for clarifying that this is a Review type of paper.

Response#2.1: Many thanks to the reviewer for providing valuable corrections and suggestions. Yes, and this review paper, will lay strong foundation for new researchers, practitioners and even specialists to know new research work belonging to the related terminologies.

---------------------------------------------------------------------------------------------------

Comment# 2.2: However, the existence of a review paper should be justified, when compared with similar previous publications.

Response#2.2:

Thank you for the comments. We have try our best to present each of the section 2,3,4,5 with fundament concepts on IoT sensor data along with latest references. The entire paper is revised with more information and even new references existed till date matching the central theme are also cited and added to related works and varied sections of the paper.

Comment# 2.3: Are there previous review works on the same topic?

Response#2.3: As of our study analysis and knowledge, there exists no single work that provide complete overview of these IoT sensor data concepts. And this will be the first comprehensive paper with regard to IoT Sensor Data Processing, Fusion and Analysis Techniques.

--------------------------------------------------------------------------------------------------

Comment# 2.4: Is this review its first of its kind?

There are several works related to the particular problems such as IoT data analysis, IoT data fusion, and IoT convergence with emerging technology do exist in the literature. However, as per our knowledge, our paper is first of its kind to provide complete perspective of IoT Sensor data. We have even done strong search on varied global databases and we have not found as such single paper matching to the central theme and matching the core domain related to our paper.

Response#2.4:

---------------------------------------------------------------------------------------------------

Comment# 2.5: Does the proposed review paper offers a more complete, or updated analysis, or with a different emphasis from previous reviews?

Response#2.5: Yes, our paper provides complete and updated analysis as of 2020. We have updated paper to the best possible and till date matching related works.

---------------------------------------------------------------------------------------------------

Comment# 2.6: Some comments could be incorporated in the introduction in order to highlight better the contributions in terms of review content.

Response#2.6: Thank you. We have updated the manuscript introduction section as suggest by the reviewer.

---------------------------------------------------------------------------------------------------

Comment# 2.7: Authors could consider the following references if they offer similar, overlapping themes:

  • Djedouboum, A.C.; Abba Ari, A.A.; Gueroui, A.M.; Mohamadou, A.; Aliouat, Z. Big Data Collection in Large-Scale Wireless Sensor Networks. Sensors 2018, 18, 4474.
  • Abu-Elkheir, M.; Hayajneh, M.; Ali, N.A. Data Management for the Internet of Things: Design Primitives and Solution. Sensors 2013, 13, 15582-15612.
  • Cai, B. Xu, L. Jiang and A. V. Vasilakos, "IoT-Based Big Data Storage Systems in Cloud Computing: Perspectives and Challenges," in IEEE Internet of Things Journal, vol. 4, no. 1, pp. 75-87, Feb. 2017, doi: 10.1109/JIOT.2016.2619369.
  • Ma, P. Wang and C. Chu, "Data Management for Internet of Things: Challenges, Approaches and Opportunities," 2013 IEEE International Conference on Green Computing and Communications and IEEE Internet of Things and IEEE Cyber, Physical and Social Computing, Beijing, 2013, pp. 1144-1151, doi: 10.1109/GreenCom-iThings-CPSCom.2013.199.

Response#2.6:

Thank you for providing details of these references. As suggested by the reviewer, we have referred most of the relevant papers and cited them appropriately within the manuscript. And we are highly thankful for bringing these related works to our knowledge for suitable incorporation to our manuscript.